# Preparation and Mechanical Properties of PBAT/Silanized Cellulose Composites

Xiangyun Wang [1,2], Wenlong Mo [3], Yongming Zeng [1,2,4,*] and Jide Wang [3,*]

1   Department of Chemistry and Chemical Engineering, Changji University, Changji 831100, China; xyunwangai@163.com
2   Xinjiang Key Laboratory of High Value Green Utilization of Low-Rank Coal, Changji 831100, China
3   Department of Chemistry and Chemical Engineering, Xinjiang University, Urumqi 830000, China; mowenlong@xju.edu.cn
4   School of Chemistry and Chemical Engineering, Nanjing University, Nanjing 210023, China
*   Correspondence: zym903@126.com (Y.Z.); awangjd@126.com (J.W.)

**Abstract:** Polybutylene adipate-terephthalate (PBAT) is a fully biodegradable polyester, which has been proven to be the most suitable alternative to traditional plastics. However, due to the low strength of PBAT (17.2 MPa) and high price, the use of PBAT has a degree of limitations. To obtain a cost-effective and high-performance composite material of PBAT, for this study we selected microcrystalline cellulose, which is inexpensive and easily available, as the reinforcing medium. However, due to the hydrophobicity of PBAT, the mechanical properties of PBAT when mixed with hydrophilic cellulose were low. In order to improve the compatibility of cellulose and PBAT, this study selected hexadecyltrimethoxysilane (HDTMS) containing long carbon chains to silanize microcrystalline cellulose (MCC) to obtain silanized cellulose (SG). Three types of SGs with different degrees of silanization were obtained by controlling HDTMS with different mass ratios (1:10; 3:10; 5:10) to react with MCC. Characterization of these three types of SGs was conducted using FTIR, TEM, and water absorption analysis. The results demonstrated the successful synthesis of SG. With the increase in the reaction ratio of HDTMS and MCC, the size of the nanoparticles increases, and the water absorption decreases significantly. Subsequently, PBAT/SG composites were prepared by blending three kinds of silanized cellulose with PBAT in different proportions by the sol-gel method. To study the thermal stability and compatibility, the mechanical properties of the composites were evaluated, including thermogravimetric testing, scanning analysis, and dynamic thermomechanical testing. The optimal blending ratio and the optimal type of silane cellulose were found. Analysis of the mechanical properties revealed that the tensile strength initially increased and then decreased with increasing blending ratio for all three composites tested. Among them, the PBAT/SG2 composites exhibit superior performance, with a maximum tensile strength reaching 22 MPa at an 85/15 blending ratio, nearly 30% higher than that of pure PBAT alone. The addition of SG significantly improved the strength of the PBAT, and SG2 is more suitable for preparing high-strength composite materials. In addition, after the addition of SG, the yield stress of the composite is improved while maintaining good thermal stability. Both the SEM and DMA results indicated good compatibility of the PBAT/SG composites. This study provides a new idea for the industrial-scale development of degradable polyesters with low cost and good mechanical properties.

**Keywords:** silanized cellulose; sol-gel; compatibility; polybutylene adipate-terephthalate

## 1. Introduction

With the increasing severity of environmental pollution and the growing consciousness of environmental issues, an increasing amount of attention is being directed toward the advancement and utilization of biodegradable materials. Biodegradable materials, also known as "green plastics", can undergo degradation into water and carbon dioxide

through microbial activity under natural or compost conditions [1,2]. Among the various biodegradable materials, polyadipate-butylene terephthalate (PBAT) is the most extensively employed and promising option. PBAT is flexible and rigidly soluble in aromatic polyesters, making it a highly suitable substitute for conventional plastics [3–5].

However, the high production cost and low intensity of PBAT have emerged as significant obstacles impeding its widespread application as a potential substitute for polyethylene and other plastics [6]. In recent years, numerous studies have focused on enhancing the intensity of PBAT. The commonly employed approach involves blending PBAT with high-strength substances, including high-strength polymers, rigid inorganic particles, and cellulose [7–10].

Among the many polymers blended with PBAT, polylactic acid (PLA) has become one of the most popular high-strength polymers due to its excellent mechanical and processing properties [11]. Yeh [12] prepared a PBAT/PLA composite material by melt blending and reported that when the PBAT content was low, the PBAT could be evenly dispersed in the PLA matrix, and the tensile strength of the composite improved. When the content of PBAT is above 5%, phase separation of the composite system also occurs, and the tensile strength decreases significantly. To improve the compatibility of PBAT and PLA, Chen [13] used a small amount (0.5–3 wt%) of epoxy-functional styryl-acrylic oligomer (ESA) as a high-efficiency crosslinking agent to prepare super toughened and mechanically robust PLA/PBAT blends by dynamic vulcanization. When ESA was added to the surface, the PBAT phase had a strong interfacial adhesion force, which gave rise to the highest impact toughness and ductility while maintaining high strength.

The inorganic particles used to improve the strength of PBAT include carbon nanotubes and $CaCO_3$. The mechanical properties of carbon nanotubes (CNTs) are excellent, with a theoretical strength of 150 GPa and a density of only 1/6 that of steel [14]. The use of CNTs to modify polymers has obvious advantages [15]. Zhao [16] added carbon nanotubes (CNTs) with different contents to PBAT/PLA blends to form branched-chain carbon nanotube copolymers using multifunctional epoxy oligomers (ADRs) as reaction compatibilizers. The results showed that the addition of CNTs and ADR improved both the strength and toughness of the samples. The impact strength was 35.3 kJ/m$^2$, approximately 7 times that of the PLA/PBAT blend, and the tensile strength increased from 33.6 MPa to 42.8 MPa. The performance of the PLA/PBAT blends co-modified with ADR and CNTs was significantly better than that of the PLA/PBAT blends co-modified with ADR or CNTs. $CaCO_3$ has strong toughening and strengthening effects and can significantly improve the bending strength and bending modulus of materials [17] and enhance their thermal stability. Liu [18] designed and prepared PBAT/$CaCO_3$ composite films with PBAT as the resin matrix and calcium carbonate ($CaCO_3$) as the filler by using a twin-screw extruder and a single screw extrusion blow molding machine. The results showed that the size and content of the $CaCO_3$ particles significantly influence the tensile properties of the composites. The addition of unmodified $CaCO_3$ reduces the tensile properties of the composites by more than 30%. The modification of $CaCO_3$ by the titanate coupling agent 201 (TC-2) improved the overall performance of the PBAT/$CaCO_3$ composite film. When the addition of TC-2 was 1%, the maximum tensile strength of the film was 20.55 MPa, the water vapor permeability of the composite was reduced by 27.99%, and the water vapor permeability coefficient was reduced by 43.19%.

Cellulose has unique properties, such as low density, high toughness, high strength [19,20], and complete degradation. In addition, cellulose comes from a wide range of sources and is inexpensive, so the price of cellulose-based composites is relatively low [21,22]. However, cellulose also has several drawbacks, mainly because it is a polyhydroxyl compound that absorbs water, and if it is directly blended with hydrophobic polyester, an incompatible phenomenon will occur [23,24]. As a result, the properties of composite materials cannot be optimized, which restricts the use of cellulose to a certain extent. Giri [25] prepared microcrystalline cellulose (MCC) from wheat straw as a raw material and prepared the composite material by melting composites with PBAT in different proportions. The results

showed that the composite material could maintain good mechanical properties when the MCC filling amount was low, and when the MCC filling amount exceeded 10%, the composite material could maintain good mechanical properties. Due to the aggregation of MCC and poor interfacial bonding, the PBAT composite cracks prematurely and leads to fracture when subjected to external forces. In order to improve the compatibility of cellulose and PBAT and to improve the properties of composite materials, it is necessary to modify cellulose. At present, the modification methods reported in the literature are mainly chemical modification. Hou [26] used octadecylamine (ODA) to graft nanocellulose (CNF) to improve its compatibility with the PBAT matrix. PBAT composites containing 1 wt% CNF were prepared by the masterbatch premixing method to avoid CNF aggregation during extrusion. The results showed that the tensile strength of the CNF(OCNF)/PBAT-fused extrusion composite was 17.2% greater than that of the PBAT polymer without affecting the thermal stability of the PBAT. To enhance the compatibility between nanocellulose and PBAT, Morelli [27] used 4-phenylbutylisocyanate to modify cellulose nanocrystals (CNC) and prepared PBAT/CNC composites by melting extrusion. The results showed that the dispersion of CNC in the PBAT matrix was greatly improved after modification. The composite, with 5% modified CNC, had a consistent appearance compared to pure PBAT, however, the unmodified composites had obvious brown spots. The elastic modulus of the modified CNC composite was increased by 55%, and the water vapor transmittance was decreased by 63%.

At present, there have been many studies dedicated to increasing the mechanical properties of PBAT, but there are several problems that remain to be addressed, such as the limited strength improvement and poor compatibility of blended materials. To reduce the cost of the material, improve the compatibility of PBAT and additives, and further improve the mechanical properties of the material, for this study we selected inexpensive and easily available microcrystalline cellulose as the strengthening medium and selected hexadecyl trimethoxysilane, which contains a long carbon chain, to silanize the microcrystalline cellulose to obtain silanized cellulose (SG). PBAT/SG composites were then blended with PBAT to prepare PBAT/SG composites, and the thermal stability, compatibility, and mechanical properties of the composites were further studied through thermogravimetric testing, scanning analysis, dynamic thermomechanical property analysis, and mechanical property analysis to explore the mechanism of enhancing the mechanical properties of the composites. This approach provides a new insight for further study of the blending of degradable polyesters.

## 2. Materials and Methods

### 2.1. Sources of the Materials

The PBAT, provided by Xinjiang Lanshan Tunhe Co., Ltd. (Urumqi, China), exhibits a number average molecular weight of $5 \times 10^4$, a polymer dispersion index of 1.7, a resin density of approximately 1.21 g/cm$^3$, and a melting index of 3–5 (g/10 min at 190 °C). Microcrystalline cellulose, column chromatography, Shanghai Hengxin Chemical Reagent Co., Ltd. (Shanghai, China); sodium hydroxide, analytically pure, Tianjin Zhiyuan Chemical Reagent Co., Ltd. (Tianjin, China); thiourea, analytically pure, Tianjin Shengmiao Chemical Reagent Co., Ltd. (Tianjin, China); cetyltrimethoxysilane, analytically pure, Shanghai Aladdin Co., Ltd. (Shanghai, China); anhydrous ethanol, analytically pure, Tianjin Yongsheng Fine Chemical Co., Ltd. (Tianjin, China); concentrated hydrochloric acid, analytically pure, Tianjin Kemeng Chemical Plant (Tianjin, China).

### 2.2. Synthesis of SG

A total of 3 g of MCC was added to 100 g of alkali solution mixed with sodium hydroxide (NaOH) and thiourea (CH$_4$N$_2$S). (NaOH:CH$_4$N$_2$S:H$_2$O = 9.5:4.5:86 (*w/w/w*)) was uniformly stirred for 100 min to form a homogeneous solution, and the solution was frozen in a refrigerator at −20 °C for 24 h. After thawing, 0.1 g of cetyltrimethoxy-silane

(HDTMS) was added to 100 g of cellulose solution, and the mixture was stirred evenly to form a sol. Then, 15 mL of 4 mol/L HCl was added until a gel was formed.

Silanized cellulose 1 (SG1) was obtained by aging the gel at room temperature for 10 h, washing it with distilled water to a neutral pH, and drying it to constant weight in a drying oven at 100 °C. The preparation methods for silanized cellulose 2 (SG2) and silanized cellulose 3 (SG3) were the same as above, except that the mass ratio of HDTMS and MCC were 3:10 and 5:10, respectively.

### 2.3. Preparation of PBAT/SG Composites

SG was ground into 200–300 mesh powder and dried with PBAT in a blast drying oven at 70 °C for 12 h. The solution blending method [28,29] was used to mix SG and PBAT evenly; that is, PBAT was first added to the chloroform solution and evenly stirred until all was dissolved. Then, SG was added, and the mixture was continuously stirred until SG was evenly dispersed in the PBAT solution. After the solvent volatilized completely, the composite material was extruded on a twin-screw extruder (L/D = 40:1, HAAKE, Karlsruhe, Germany) at 130 °C for granulation, and reinjected into standard splines. The process flow chart see Figure 1 below:

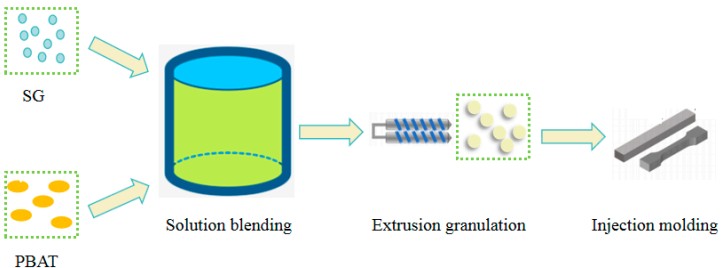

**Figure 1.** Preparation process for the PBAT/SG composites.

### 2.4. Measurements and Characterizations

Before the performance test, the sample was treated to eliminate the residual internal stress of the composite material and remove moisture. The method was as follows: The composite material was placed in a blast box at 80 °C and dried for 12 h to fully remove moisture and eliminate internal residual stress. The composite material was then stored in a sealed bag to prevent moisture absorption.

Fourier transform infrared (FTIR) spectroscopy: the potassium bromide tablet method was used to observe the chemical composition of the SGs in the range of 4000~400 cm$^{-1}$ to determine whether silanization modification occurred in the microcrystalline cellulose.

Transmission electron microscopy (TEM): SGs were dispersed in anhydrous ethanol and observed on a Hitachi H-600 transmission electron microscope (Tokyo, Japan) at an acceleration voltage of 100 kV to determine the size and dispersion of the silica in the SGs. TEM was used to analyze the composite materials via the ultrathin slice method. Tensile properties: The tensile strength of the composite was tested according to GB/T1040-1992. The drawing rate was 10 mm/min, and the average value was measured 5 times for each sample.

Scanning electron microscopy (SEM): the composite material was soaked in liquid nitrogen for 40 min, followed by brittle fracture, gold spraying on its cross section, and observation of the cross-section morphology of the composite material by scanning electron microscopy at an accelerating voltage of 15 kV.

Thermogravimetric analysis (TGA): A sample of the PBAT/SG composite material was cut into small pieces of approximately 5 mg, and the temperature was increased to 1000 °C at a heating rate of 10 °C/min under a nitrogen atmosphere. The heating process data were recorded, and thermogravimetric analysis of the composite material was performed.

Dynamic thermal analysis (DMA): A Q800 V7.5 DMA (Boston, MA, USA) tester was used for dynamic thermal mechanical testing of the composite materials. The temperature

range was −50–110 °C, the heating rate was 3 °C/min, and the instrument was operated in strain mode at a fixed frequency of 1 Hz. The whole test process was carried out in a $N_2$ atmosphere.

Mechanical property: The tensile strength of the composite was measured at room temperature using a CMT6104-50N electronic universal testing machine manufactured by Shenzhen Xinsi Measurement Co., Ltd. (Shenzhen, China). The drawing rate was 10 mm/min. At least 5 valid data were collected for each group of samples to calculate the mean value and standard deviation.

## 3. Results and Discussion

### 3.1. Synthesis and Characterization of SG

(1) Sol-gel reaction

SG is prepared by the sol–gel method, which includes the hydrolysis and condensation of HDTMS in cellulose solution. When HDTMS is added to an alkali solution of cellulose, hydrolysis occurs, the three methoxy groups of HDTMS are hydrolyzed into hydroxyl groups, and methanol is obtained as a byproduct. Methanol helps HDTMS dissolve in water [30] and therefore facilitates the hydrolysis reaction. When HCl was added to the solution, the condensation rate increased. After hydrolysis, part of the hydroxyl group of HDTMS condenses with the hydroxyl group of MCC to form a Si-O-C covalent bond, and part of the hydroxy group is self-condensed to form a polysiloxane network structure. The condensation reaction of HDTMS and MCC results in cross-linking between microcrystalline cellulose, resulting in the hydrophobicity of silanized cellulose. Synthetic route of silanized cellulose see Figure 2 below:

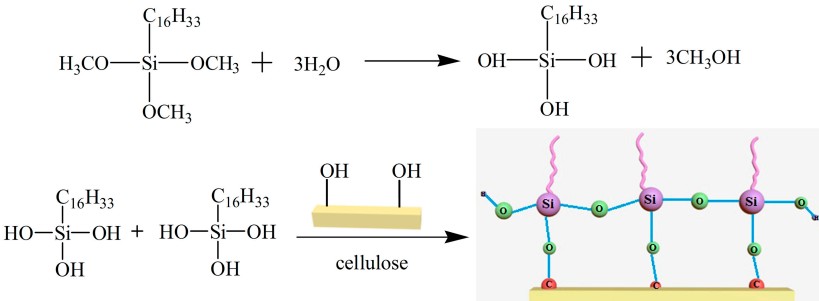

**Figure 2.** Synthetic route of silanized cellulose.

(2) IR and TEM analysis of SGs

As seen from the infrared spectrum of MCC (Figure 3a), the peak attributed to the stretching vibration of -OH groups ranged from 3600 to 3200 cm$^{-1}$. The strong absorption peak near 1056 cm$^{-1}$ is the characteristic peak of C-O-C in hemiacetal pyran sugar. The absorption peak of C-H is at approximately 2898 cm$^{-1}$. As the ratio of HDTMS to MCC increases, the absorption peak attributed to OH in SG becomes increasingly weaker (Figure 3b–d), mainly because MCC condensation occurs during the silanization of cellulose and possibly because Si-OH, like the OH in MCC [31], can also form intramolecular hydrogen bonds in its own chains. Compared with that of MCC, the absorption peak band corresponding to C-H is more intense at 2840–2930 cm$^{-1}$ for SG. With the increase in the reaction proportion of HDTMS and MCC, the corresponding peak intensity also significantly increases [32], which further proves that the sol-gel reaction introduces alkyl chains. However, it is difficult to determine Si-O-Si and Si-O-C on SG in the spectrum [33] because the absorption peaks of Si-O-Si and Si-O-C overlap near 1100–1150 cm$^{-1}$. However, the increase in the intensity of the SG absorption peak near 1100–1150 cm$^{-1}$ also indirectly proves that the sol-gel reaction occurred.

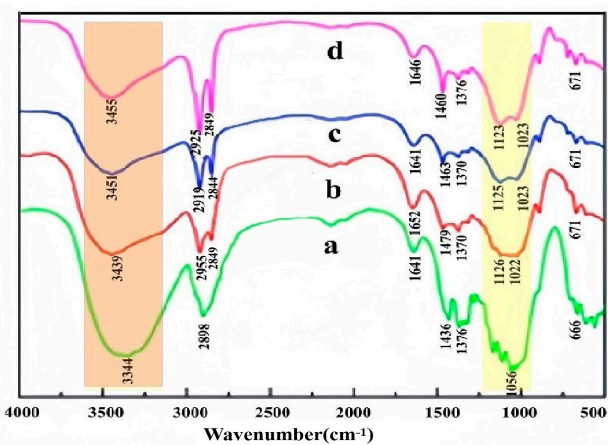

**Figure 3.** FTIR spectra of MCC (a) and silanized celluloses with different mass ratio of HDTMS to MCC (1:10 (b); 3:10 (c); 5:10 (d)).

The TEM image in Figure 4a,b,d shows that after the sol-gel reaction, $SiO_2$ nanoparticles are formed in the SG, and the size of the nanoparticles increases with increasing reaction proportion of HDTMS and MCC, whose sizes have increased from 11 nm to 21 nm, and the average particle size of nano-silica in SG2 is 17.32 nm.

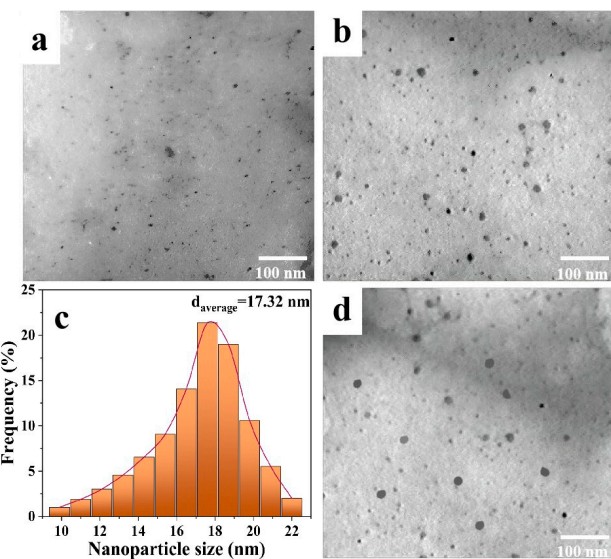

**Figure 4.** TEM images: silanized celluloses with different mass ratio of HDTMS to MCC (1:10 (**a**); 3:10 (**b**); 5:10 (**d**)); and (**c**) average diameter of silanized celluloses with mass ratio of HDTMS to MCC (3:10).

(3)  Water absorption of SG

The wettability of solids is usually measured by the water contact angle, but due to the uneven surface of SGs, it is difficult to detect the baseline water droplets [34]. Therefore, in this study, the degree of silanization and hydrophobicity of SGs were measured by the amount of water absorbed. Before silanization, MCC has high water absorption capacity and can be almost completely wetted by water. With an increasing reaction ratio of HDTMS to MCC, the water absorption rate decreased significantly, and the water absorption rate of SG3 was only 0.54% (Figure 5). When HDTMS modifies MCC, on the one hand, the hydroxy groups of cellulose and silanol undergo dehydration, which causes the long hydrophobic carbon chain of HDTMS to bind to MCC; on the other hand, the intramolecular condensation reaction results in the formation of a network structure, which also reduces the water absorption of SG.

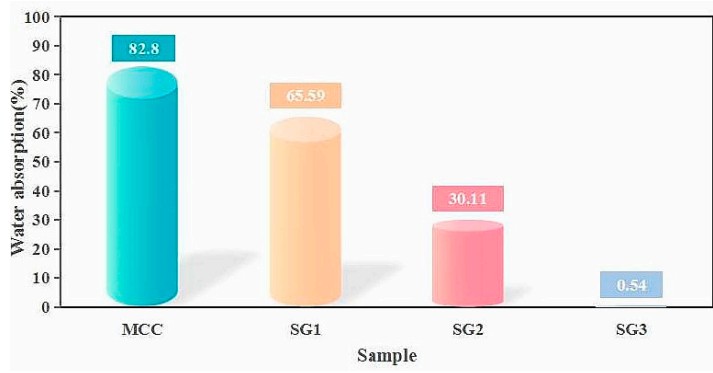

**Figure 5.** Water adsorption of MCC and SG with different mass ratio of HDTMS to MCC.

### 3.2. Performance Analysis of Composite Materials

### 3.2.1. IR and TEM Analysis of Composite Materials

Samples of PBAT/SG blend materials were tested by FTIR and TEM to analyze the changes in the composite groups after blending PBAT and SG and the dispersion of SG in PBAT. The results are shown in Figure 6.

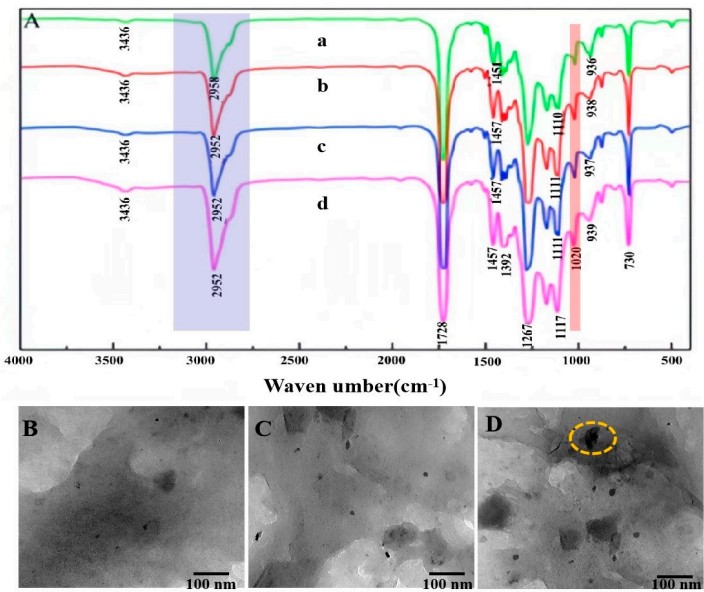

**Figure 6.** IR and TEM images of (a and (**A**)) PBAT, (b and (**B**)) PBAT/SG1, (c and (**C**)) PBAT/SG2, and (d and (**D**)) PBAT/SG3.

PBAT belongs to the polyester family, and the stretching vibration peak of C-O near 1111 cm$^{-1}$ and the stretching vibration peak of C=O near 1728 cm$^{-1}$ are the characteristic peaks of esters [35]. C-H asymmetric and symmetric bending vibration absorption peaks are found at approximately 1457 cm$^{-1}$ and 1392 cm$^{-1}$, respectively. The stretching vibration peak of -OH was near 3436 cm$^{-1}$ (Figure 6A).

Compared with those of pure PBAT, the positions of the characteristic absorption peaks of the composite material are basically unchanged, but the intensities of some absorption peaks change obviously. The strength of the peak attributed to the stretching vibration of the composite material becomes stronger near 1020 cm$^{-1}$. This is because of the stretching vibration peaks of C-O-C and Si-O-C formed by SG near 1020 cm$^{-1}$. The intensity of the C-H absorption peak at 2952 cm$^{-1}$ was significantly greater than that of the other peaks, especially for PBAT/SG2 and PBAT/SG3. This was because SG2 and SG3 had the highest degree of silanization and contained more alkanes than SG1.

Figure 6B–D shows TEM images of PBAT/SG1, PBAT/SG2, and PBAT/SG3, respectively. The figure shows that the SG particle size gradually increases, but the dispersion

in PBAT is mainly uniform. However, the agglomeration of silica particles in Figure 6D is probably caused by the large size of silica particles.

### 3.2.2. Influence of the SG Dosage on the Mechanical Properties of the Composite Materials

Figure 7 shows the influence of the amount of SG on the tensile properties of the composites. The figure shows that the tensile strength of PBAT/SG has no obvious improvement trend when the dosage of SG is low; this is mainly because when the amount of SG added is low (when the blend ratio is approximately 95/5), the continuous phase structure of the base resin PBAT is destroyed, causing defects in the composite system, resulting in stress concentration during tensile testing and a slight reduction in tensile strength. When the amount of SG continues to increase, the tensile strength of the composite improves to a certain extent, which is due to the interaction between SG and the PBAT matrix through hydrophobic and polar interactions, etc., which enhances the compatibility of the composite material, and the combination of inorganic silicon particles also enhances the tensile strength. However, when the amount of SG is too high, on the one hand, the enhanced interaction between SG and the PBAT matrix restricts the movement of PBAT chain segments; on the other hand, due to the grain distribution of SG, both the increase in the SG content and the trend toward phase separation increase, which causes the mechanical properties of the composite to decline.

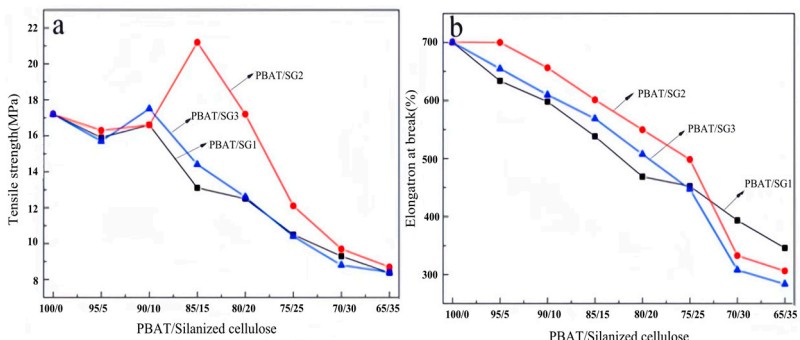

**Figure 7.** Tensile strength and elongation at break of composites with different SG amounts. (**a**) tensile strength of composites; (**b**) elongation at break of composites.

In addition to the amount of SG, the tensile strength is also affected by the size and hydrophilicity of the SG. Among the three composites, PBAT/SG1 and PBAT/SG3 both reach the maximum tensile strength when the blending ratios of PBAT with SG1 and SG3 are 90/10, which are 16.4 MPa and 17.3 MPa, respectively. Compared with those of the previous two materials, the mechanical properties of PBAT/SG2 are better. When the blending ratio of PBAT to SG2 is 85/15, the tensile strength of the composite material reaches 22.0 MPa, which is nearly 30% higher than pure PBAT. This is because SG1 has strong hydrophilicity and poor interface compatibility with PBAT, which cannot effectively improve the mechanical properties of PBAT. Although the hydrophobicity of SG3 is very good, due to the large particles, agglomeration easily occurs, and the mechanical properties of PBAT cannot be effectively improved. In contrast, SG2 is not only hydrophobic but also has a moderate and uniform particle size, which can effectively improve the mechanical properties of PBAT. Li [36] prepared K-cellulose by γ-(2, 3-epoxypropoxy) propytrimethoxysilane (KH560) modified cellulose and found that the mechanical properties of the composite were the best when 2% K-cellulose was added, with a maximum tensile strength of 18.3 MPa, an elongation at break reduced by about 21%, and a tensile strength about 17% lower than that in this study. In addition, the low addition amount of K-cellulose is not conducive to the reduction of the cost of composite materials.

The elongation at break of the three composites gradually decreased with increasing SG concentration. However, before the blending ratio reached 75/25, the elongation at break was the highest for PBAT/SG2, followed by that for PBAT/SG3, and that for PBAT/SG1

was the lowest. The elongation at break of PBAT/SG2 and PBAT/SG3 decreased rapidly, while the elongation at break of PBAT/SG1 did not decrease significantly. The introduction of SG isolates the interaction between PBAT molecular chains and increases the flexibility of molecular chains; on the other hand, silica plays a role as a filler, which limits the conformational change of molecular chains. Therefore, at a certain blending ratio, the hydrophobic effect of SG plays a leading role, and the good compatibility of the resulting material shows that the elongation at break of PBAT/SG2 is high. However, with a further increase in the SG proportion, the limiting effect of SG as a filler becomes more dominant, so the elongation at break decreases rapidly for the SG2 and SG3 series with larger particle sizes.

Figure 8 shows the stress–strain curve of the PBAT/SG composites. The slope of the linear part of the stress–strain curve is used as the dynamic elastic modulus of the sample. It can be seen from the figure that the elastic modulus of the composite materials is higher than that of PBAT. Among them, the elastic modulus of PBATSG2 is the largest, followed by PBATSG3, and PBATSG1, which is mainly because cellulose molecules have a large number of hydroxyl groups, and the strong intramolecular and intermolecular hydrogen bonding force enhances the rigidity of the molecule, so its rigidity is significantly higher than that of the flexible PBAT. Due to the additive modulus of elasticity, the modulus of the rigid modified cellulose after adding PBAT is greater than that of PBAT, and the amount of SG2 (15%) in PBAT/SG2 composites is higher than that of SG1 and SG3 (10%), so the elastic modulus of PBAT/SG2 is the largest. The elastic modulus of PBAT/SG3 is greater than that of PBAT/SG1, probably because the rigidity of cellulose is improved by silanization. Elastic deformation occurs when the strain is 33.9%. When the yield stage is reached, the yield stress of the composites is greater than that of the PBAT. The yield stress of PBAT/SG2 is the highest (10.7 MPa), followed by that of PBAT/SG3 and PBAT/SG1. This is related to the elastic modulus of the material. After the yield stage, the deformation generated by the material is irreversible permanent deformation. When the stress reaches the highest point, the material is pulled off, and the maximum stress decreases in the order of PBAT/SG2 > PBAT/SG3 > PBAT/SG1 > PBAT. The elongation at break values of the PBAT, PBAT/SG1, PBAT/SG2, and PBAT/SG3 composites are 657.2%, 558.9%, 564.2%, and 574.8%, respectively, indicating that the toughness of the composites is very good.

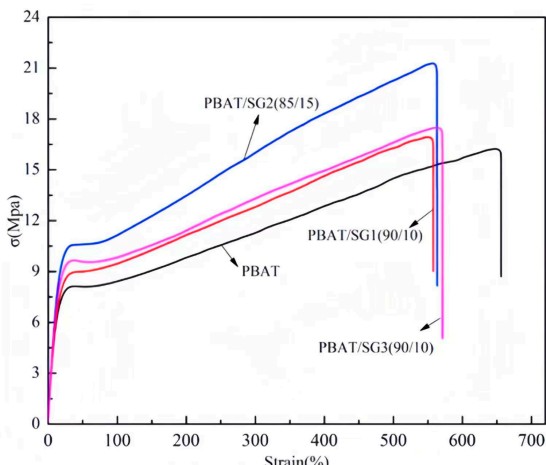

**Figure 8.** Stress–strain curves of the PBAT/SG composites.

Figure 9 shows the corresponding states of splines of PBAT, PBAT/SG1(90/10), PBAT/SG2(85/15), and PBAT/SG3(90/10) before stretching and after fracture.

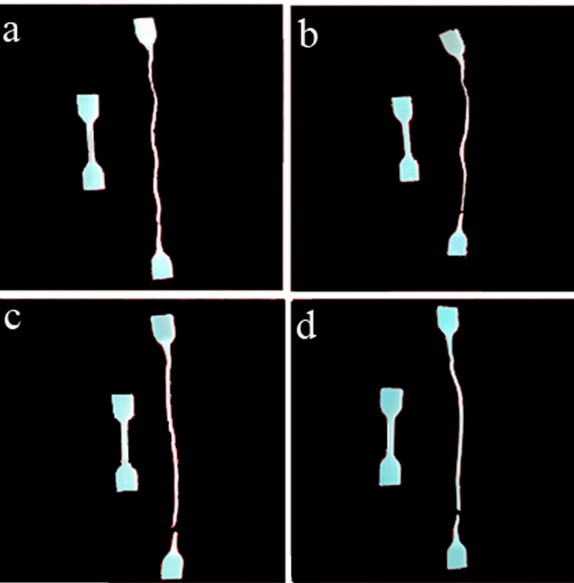

**Figure 9.** Photos before and after break of (**a**) PBAT, (**b**) PBAT/SG1 (90/10), (**c**) PBAT/SG2 (85/15), and (**d**) PBAT/SG3 (90/10).

### 3.2.3. Thermal Behavior Analysis of Different Kinds of Composite Materials

The thermal properties of PBAT and its three composites are shown in Figure 10. It can be obtained from Figure 10A that the thermal decomposition of PBAT and the PBAT/SG composites begins at approximately 320 °C. In contrast, the initial degradation temperature of the composite is slightly lower than that of PBAT, which may be caused by the low degradation temperature of the alkyl chain in SG. The weight loss of PBAT and its composite materials mainly occurs in the range of 350–430 °C. When the temperature reaches 1000 °C, almost no PBAT remains, while the residual amounts of PBAT/SG1, PBAT/SG2, and PBAT/SG3 are 0.91%, 3.2%, and 2.9%, respectively, which are the $SiO_2$ generated by SG oxidation at high temperature.

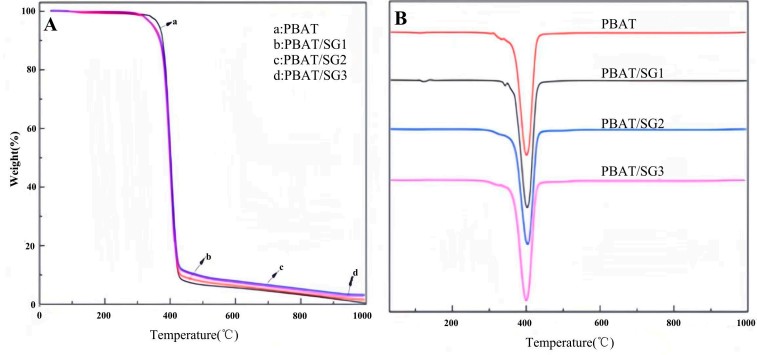

**Figure 10.** TGA (**A**) and DTG (**B**) curves of (a and **A**) PBAT, (b and **A**) PBAT/SG1 (90/10), (c and **A**) PBAT/SG2 (85/15), and (d and **A**) PBAT/SG3 (90/10).

In Figure 10B, PBAT and the three composites have only one peak, which is caused by the degradation of PBAT and the three composites. The thermal degradation temperatures of these four materials are similar, indicating that the addition of silanized cellulose has little effect on the thermal properties of the composites.

### 3.2.4. SEM Analysis of Different Kinds of Composite Materials

The cross-sectional morphology of the composite was observed via SEM, and the compatibility of the two phases of the composite was further studied. Figure 11 shows the SEM image of the brittle liquid nitrogen section of PBAT and its three composites. The cross

section of the PBAT is relatively flat, indicating that the PBAT underwent brittle fracture after being treated with liquid nitrogen (Figure 11a). There was little difference in the cross-sectional morphology of the three composites (Figure 11b–d). The cross sections of the PBAT/SG1, PBAT/SG2, and PBAT/SG3 composites are all flat, indicating that SG1, SG2, and SG3 have good dispersion in PBAT. This is because the silanization modification causes the hydrophilic hydroxyl group on the cellulose molecular chain to be replaced by a hydrophobic Si-O-C bond, which improves the interdependence of the two phases, so the surface of the composite material is flat. However, only the surface of PBAT/SG3 composite has holes and insignificant particles, because the silica nanoparticles in SG3 are larger in size and agglomerate to form larger particles, and thus separate from the matrix during the mixing process.

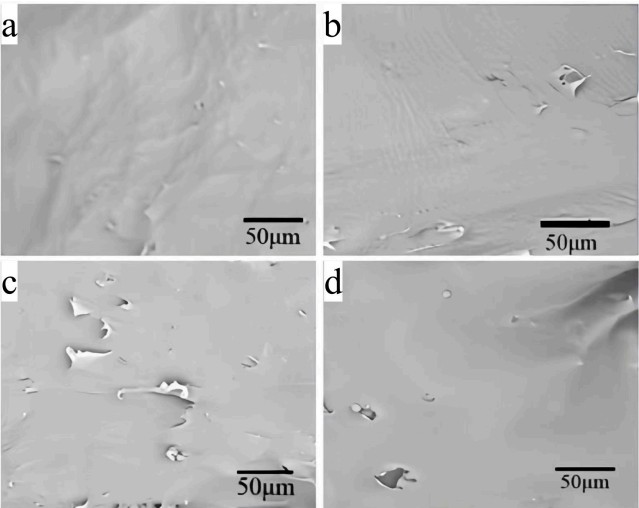

**Figure 11.** SEM images of (**a**) PBAT, (**b**) PBAT/SG1 (90/10), (**c**) PBAT/SG2 (85/15), and (**d**) PBAT/SG3 (90/10).

### 3.2.5. DMA Analysis of Different Kinds of Composite Materials

The DMA curve of PBAT/SG provides information on not only the molecular chain movement of the composites, but also the glass transition temperature (Tg) of the PBAT and composites. In addition, the modulus variation rule of composites can be obtained, which is highly important for determining the dynamic mechanical properties of composites.

Figure 12 shows the curves of the energy storage modulus (A), damping factor (B), and loss modulus (C) of the PBAT and its composite material with changing temperature. In the figure, the sharp decrease in the energy storage modulus and the peak damping factor correspond to the glass transition temperature of the material. The energy storage module scale is a measure of the elasticity (rigidity) of a material and describes the energy stored by elastic deformation during the deformation process. Figure 12A shows that the energy storage modulus of PBAT and the three other composites all decrease with increasing temperature. The glass transition temperature of PBAT is $-20.7\ ^\circ$C, and the glass transition temperatures of the PBAT/SG1 and PBAT/SG3 composites increase slightly. The glass transition temperature of the PBAT/SG2 composites decreased slightly to $-22.1\ ^\circ$C and $-15.8\ ^\circ$C. The results indicated that the blending of SG and PBAT, on the one hand, had a certain plasticizing effect, promoted the movement of polymer molecular segments, and thus reduced the Tg. On the other hand, the introduction of rigid silica particles increases the Tg. Therefore, the blending ratio of SG and PBAT, as well as the size of the silica particles, will affect the Tg.

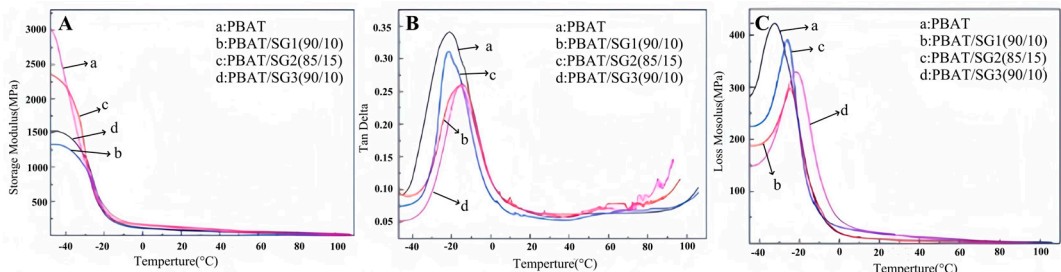

**Figure 12.** DMA curves of storage modulus (**A**), damping factor (**B**), and loss modulus (**C**) of PBAT and composites at different temperatures.

In addition, the energy storage modulus of PBAT is the highest at low temperature. After blending, the modulus of the composite materials is lower than that of PBAT. Among the three composite materials tested, PBAT/SG2 has the highest modulus, 2362.3 MPa, followed by PBAT/SG3 and PBAT/SG1, 1540.2 MPa and 1334.7 MPa, respectively. Since the energy storage modulus is a measure of the elasticity of the material, among the three composites, PBAT/SG2 has the best elasticity, which is consistent with the elastic modulus results.

Figure 12B shows the curves of the relationship between the damping coefficient and temperature of PBAT and its composite. In the figure, all three composites have only one damping loss peak, indicating good compatibility between PBAT and SG, which is consistent with the above SEM results. Figure 12C shows the curves of the relationship between the loss modulus and temperature of PBAT and its composite. In the figure, the peak value of PBAT is larger than that of the other three composites. In the blending system, the loss modulus of PBAT/SG2 is the largest, at 392.0 MPa, followed by that of PBAT/SG3, and the loss modulus of PBAT/SG1 is the smallest, at 292.8 MPa, indicating that the addition of SG reduces the internal friction of polymer molecules and the relative loss.

## 4. Conclusions

In this work, MCC and HDTMS with long carbon chains were silanized to obtain SGs, and three SGs with different silanization degrees were obtained by controlling the reaction ratio. By the sol-gel method, three SGs were mixed with PBAT at different blending ratios to prepare the PBAT/SG1, PBAT/SG2, and PBAT/SG3 composites. PBAT/SG composites have only one glass transition temperature, and the cross section of the composite is flat, which indicates that the compatibility of these two phases is good. The thermogravimetric test results showed that the yield stress of the composites increased while maintaining good thermal stability. The test results of the mechanical properties showed that the tensile strength of the three composites first increased and then decreased with increasing blending ratio. When the blending ratios of the PBAT/SG1, PBAT/SG2, and PBAT/SG3 composites are 90/10, 85/15, and 90/10, respectively, the tensile strength reaches the maximum value. The elongations at break were 16.4 MPa, 22.0 MPa, and 17.3 MPa, while the elongations at break were 601.7%, 577.6%, and 592.4%, respectively. Among the three composites, the composite with the best performance was PBAT/SG2. When the blending ratio is 85/15, the tensile strength is nearly 30% greater than that of pure PBAT, mainly because SG2 not only is hydrophobic but also has a moderate and uniform particle size coupled with good compatibility, which can effectively improve the mechanical properties of PBAT. It is also higher than the results of modified cellulose reinforced PBAT that have been reported so far, which fully indicates that silanized cellulose is a good reinforcing material. In addition, the composites prepared in this study have high low-cost cellulose content, which can effectively reduce the cost of materials, so this study provides a green and feasible approach for the study of low-cost and high-strength PBAT. More importantly, PBAT/SG composite, as a fully biodegradable composite, has the potential to be used as an alternative to traditional commodity polymers, which can reduce the use of non-biodegradable plastics and alleviate environmental problems.

**Author Contributions:** J.W. and Y.Z. developed the original idea and designed the experiments; Y.Z. directed the project; X.W. and W.M. performed the experiments; X.W. and Y.Z. co-wrote the manuscript. All the authors discussed the results and revised the manuscript. All authors have read and agreed to the published version of the manuscript.

**Funding:** This work was supported by the Natural Science Foundation of Changji University (Grant Nos. KYLK016).

**Data Availability Statement:** All data used to support the findings of this study are included within the article.

**Conflicts of Interest:** The authors declare no competing financial interest.

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
