# Peer review of "Preparation and Mechanical Properties of PBAT/Silanized Cellulose Composites"

_processes, doi:10.3390/pr12040722_

Round 1
Reviewer 1 Report
Comments and Suggestions for Authors
General comment.
Almost all experimental methods are described extremely briefly. I strongly recommend that the authors provide information obtained from experimental data in more detail.
For example:
1. The authors wrote that the “As the ratio of HDTMS to MCC increases, the absorption peak attributed to OH in SG becomes increasingly weaker”. Please prove it. For example, the integral intensity under the peaks can be calculated.
Changes in intensities are extremely difficult to compare when the curves are separated on an intensity scale. This remark also applies to Fig.4 IR.
2. The sizes of SiO2 NPs calculated as 11 nm, 17 nm and 21 nm from TEM images. How these values were obtain. What about the errors (or polydispersity)?
3. Are you sure that such a minimalistic replica “Figure 4(B-D) shows TEM images of PBAT/SG1, PBAT/SG2 and PBAT/SG3, respectively. The figure shows that the SG particle size gradually increases, but the dispersion in PBAT is relatively uniform.” was worth your time and expense to conduct TEM experiments? Add information in this section.
4. How does the structure change with tensile strength? In my opinion, it will be a good idea to take some images at different values of tensile strength.
5. How does your research relate to the statement in the introduction?
“This study provides a new idea for the industrial-scale development of degradable polyesters with low cost and good mechanical properties.”
Please discuss this in conclusion.
Minor remarks.
1. The names of the devices on which the experiments were carried out must be indicated.
2. In Fig. 3A the letters (a) – (d) next to the curves are very invisible.
3. Line 234: Please correct 1111 cm-1
The manuscript can be published after major revision.

Comments on the Quality of English LanguageMinor editing of English language required
Reviewer 2 Report
Comments and Suggestions for Authors
The presented manuscript is devoted to the production of composite materials based on PBAT. In their review, the authors note that it is possible to use polymer, inorganic, and cellulose additives to change the mechanical properties of PBAT. Among polymers, an example is given of polylactic acid. The main problem with using fillers such as cellulose is its compatibility with the synthetic polymer matrix. In the presented study, the authors propose to use microcrystalline cellulose and hexadecyl trimethoxysilane as a composite additive. The results presented do not contain statistical data. The quality of the figures needs to be improved. In addition to the strength and elongation values, it is necessary to add data for the elastic modulus.
It is best to avoid abbreviations in keyword list.
Line 25. "SG2" - the acronym must be deciphered.
2.1 Sources of the Materials. It is necessary to indicate the characteristics of the polymers used, for example, molecular weight, degree of polymerization, etc.
Lines 127, 128. “CH4N2S:H2O = 9.5:4.5:86 (w/w/w))” - the authors indicate values for three components, but give only two formulas.
Figure 2 and 3. Authors need to structure figures and captions. The captions themselves need to be corrected!
Figure 7 can be deleted; it is not informative.
3.2.3 Thermal behavior analysis of different kinds of composite materials. The data provided for DTG are not described in the text of the manuscript.
Figure 8. The y-axis labels for DTG need to be corrected.
How do the authors relate the observed morphology data and the mechanical characteristics of the resulting systems?
Lines 328-330. It is not entirely clear how the authors relate nano-sized particles to the given morphology where the scale is 50 microns?!
Figure 10. The figure does not contain the designations - a, b and c.
Lines 346-348. Do I understand correctly that the authors give two glass transition temperatures for the PBAT/SG2 sample?
Lines 368-370. Not a completely clear statement.
The conclusions of the manuscript contain the main results achieved in the work, but in my opinion require more in-depth analysis and verification.
In general, the scientific significance of the presented manuscript is not entirely clear, and the results obtained are not unique. It is not clear how much demand such composites will have. The authors do not compare the obtained data with analogues, when from such a comparison it was possible to draw a conclusion about the outstanding mechanical characteristics of the resulting composites or their unique thermal resistance, etc.
Comments on the Quality of English LanguageI recommend that authors do some editorial work and check the terminology used, e.g. Lines 368-370. Not a completely clear statement.
Reviewer 3 Report
Comments and Suggestions for Authors
The manuscript entitled “Preparation and mechanical properties of PBAT/silanized cellulose composites” by Xiangyun Wang and coworkers presents results of experimental studies of newly synthesized composite based on polybutylene adipate-terephthalate (PBAT) with different amounts of the silanized cellulose. The experimental methods were used to describe the basic physical properties of the obtained materials and are described correctly.
The manuscript presents several interesting results but it requires some changes and providing more information before making a final decision.
1. Why did the Authors choose to use silanized cellulose? There is no description in the introduction that justifies this method. Does it provide better results than other variations of the cellulose?
2. What is the source of the following sentence: “The mechanical properties of carbon nanotubes (CNTs) are excellent, with a theoretical strength of 150 GPa and a density of only 1/6 that of steel.”?
3. What is the explanation for the double peak observed around 2900 and
2845 cm-1 in Figure 3a for b, c, and d materials?
4. What is the distribution of the size of the SiO2 nanoparticles, presented in figures 3c – 3e?
5. Figure 4 presents TEM images of the composites of PBAT with different amounts of SG. The authors claim that impurities are the SG. Why it cannot be SiO2? Why the size of the impurities is almost the same?
6. It would be better to present Figure 6 before Figure 5. It is disturbing that Authors compare data for the composites with different amounts of the SG.
7. What is the physical evidence that suggests the following sentence: “Although the hydrophobicity of SG3 is very good, due to the large particles, agglomeration easily occurs, and the mechanical properties of PBAT cannot be effectively improved.”? I have not found it directly in the manuscript…
8. The Authors claim that “… only the surface of the PBAT/SG3 composite contains insignificant particles because the SG3 particle size is 21 nm, which is larger than that of SG1 (11 nm) and SG2 (17 nm) and easy to separate from the matrix.” So is it the size of SiO2 particles or SG? It is very imprecise.
Given the mentioned problems, I cannot recommend the manuscript in its present form for publication in the “Processes” Journal.
Reviewer 4 Report
Comments and Suggestions for Authors
The study focused on creating an enhanced composite material using PBAT and SG, with three variations of SG synthesized with varying degrees of silanization. This integration aimed to address the high production cost and low mechanical properties of PBAT, ultimately working towards cost-effective, mechanically robust, and environmentally friendly degradable polyesters. The manuscript is in good shape, but there's one suggestion to further explore the reinforcement mechanism.
- In Figure 5, it's observed that PBAT/SG1 and PBAT/SG3 exhibit similar performance. To delve deeper into the mechanism of reinforcement, consider adding another sample, PBAT/SG2 (90/10), to the comparison. This would help determine if the type of SG significantly matters or if the percentage of SG is the key factor in the composite's performance.
Round 2
Reviewer 1 Report
Comments and Suggestions for Authors
The article may be published as submitted.
All comments have been taken into account.
Reviewer 2 Report
Comments and Suggestions for Authors
It is necessary to carry out editorial work with the manuscript. The pictures are of poor quality and the font size is small. The text of the manuscript must be subjected to editorial work.
Line 17. "SG1, SG2, SG3" - this part can be deleted.
Line 133. It is necessary to indicate the dimension for molecular weight.
Lines 193, 194. This sentence needs to be checked and corrected.
The quality of the spectra in Figure 3 is very poor!
Figure 4 requires increasing the font size.
Line 266. Repeat "A, a"
Lines 193, 194. This sentence needs to be checked and corrected.
